# The Role of Illness Perceptions in Dyspnoea-Related Fear in Chronic Obstructive Pulmonary Disease

**DOI:** 10.3390/jcm13010200

**Published:** 2023-12-29

**Authors:** Kylie Hill, Sarah Hug, Anne Smith, Peter O’Sullivan

**Affiliations:** 1Curtin School of Allied Health, Curtin University, Perth, WA 6102, Australia; sarah.dcosta@postgrad.curtin.edu.au (S.H.); anne.smith@exchange.curtin.edu.au (A.S.); p.osullivan@curtin.edu.au (P.O.); 2Physiotherapy Department, Royal Perth Hospital, Victoria Square, Perth, WA 6000, Australia

**Keywords:** dyspnoea, COPD, illness representations, beliefs, emotions, perceptions, health behaviours, coping strategies

## Abstract

Dyspnoea is often the most distressing symptom described by people with a chronic respiratory condition. The traditional biomedical model of neuromechanical uncoupling that explains the physiological basis for dyspnoea is well accepted. However, in people with chronic obstructive pulmonary disease (COPD), measures that are linked with neuromechanical uncoupling are poorly related to the restriction in activity during daily life attributed to dyspnoea. This suggests that activity restriction that results from dyspnoea is influenced by factors other than expiratory airflow limitation and dynamic pulmonary hyperinflation, such as the ways people perceive, interpret and respond to this sensation. This review introduces the common-sense model as a framework to understand the way an individual’s lay beliefs surrounding sensations can lead to these sensations being perceived as a health threat and how this impacts their emotional and behavioural responses. The aim is to provide insight into the nuances that can shape an individual’s personal construct of dyspnoea and offer practical suggestions to challenge unhelpful beliefs and facilitate cognitive re-structuring as a pathway to reduce distress and optimise health behaviours and outcomes.

## 1. Introduction

Dyspnoea is often the cardinal symptom reported by people living with a chronic respiratory condition, such as chronic obstructive pulmonary disease (COPD). It has been defined as “a subjective experience of breathing discomfort that consists of qualitatively distinct sensations that vary in intensity and is derived from interactions amongst multiple physiological, psychological, social and environmental factors, and may induce secondary physiological and behavioural responses [1]”. Similar to pain, dyspnoea is a noxious sensation that varies both within and between people in intensity, quality and bothersomeness [2]. When perceived as a threat, dyspnoea evokes a strong emotional response [3]. In people with COPD, a common behavioural response to avoid the distress associated with dyspnoea during daily life is to minimise physical activity [4]. When dyspnoea is perceived as having a catastrophic consequence, a shallow rapid breathing pattern is often observed (reflecting hyperarousal of the sympathetic nervous system) [5], and people are at risk of panic-spectrum psychopathology [6] and agoraphobia [7].

The traditional biomedical model proposes that the perception of dyspnoea arises from the conscious interpretation of afferent and efferent signals linked to the respiratory pump [1,8]. In people with COPD, expiratory airflow limitation results from small airway disease and parenchymal destruction created by an inflammatory response to prolonged exposure to noxious particles or gases [9]. These changes reduce the elastic recoil of the lung, and the resulting flow limitation leads to gas trapping (during expiration) that creates higher end-expiratory lung volumes [10]. This worsens with the increased ventilatory demand required during physical activity, a process known as dynamic pulmonary hyperinflation [10]. Operating at higher end-expiratory lung volumes imposes elastic and threshold loads on the inspiratory muscles whilst simultaneously shortening the inspiratory muscles and reducing their mechanical advantage. This disrupts the perceived equilibrium between afferent and efferent signals associated with ventilation and is perceived as “unrewarded inspiration” [1,10]. Although this physiological model, known as neuromechanical uncoupling, is widely supported, its capacity to explain the activity restriction that results from an individual’s lived experience of dyspnoea is limited. Notably, the impairment in forced expiratory volume in one second, a standard measure of expiratory airflow obstruction, is at best weakly associated with the activity restriction that results from dyspnoea [10] and does not differ between those who do and do not experience panic attacks due to this symptom [11]. Even changes in end-expiratory volume, which quantify dynamic pulmonary hyperinflation during exercise, explain approximately 50% of the variance in measures of exertional dyspnoea [12] and only 30% of the variance in activity restriction resulting from this symptom [13].

It is clear that the activity restriction that results from dyspnoea is dependent on factors other than consequences of expiratory airflow limitation, such as the ways people perceive and interpret this sensation. The perception of dyspnoea is the product of complex neural pathways, with earlier work showing that acute dyspnoea and even the anticipation of dyspnoea activates various brain regions, such as the midbrain, limbic and paralimbic regions, anterior insular, frontoparietal networks and amygdala [14]. These neural pathways have been hypothesised to mediate the link between memory and perceiving this sensation as a health threat, which gives rise to distress, fear or anxiety [15]. Fear of dyspnoea is a near-ubiquitous experience for people with COPD [16]. Seminal work in the 1970s and 1980s described an illness perception model (i.e., the common-sense model (CSM)) to explain how an individual’s lay beliefs surrounding sensations can lead to them being perceived as a health threat and how this impacts their emotional and behavioural responses as well as coping strategies and health outcomes [17]. This model has been revised to formally operationalise processes that were inferred in the original model. The revision also specifies moderators of model effects, such as illness type, individual differences (e.g., optimism and perfectionism) and sociostructural variables. [18]. In this narrative review, we draw on earlier work that has applied the CSM to understand beliefs and expectations around dyspnoea [19] as well as our own clinical and research experience working with people with COPD [20]. We also provide parallels of how the CSM has been used to understand another noxious symptom: chronic low back pain [21,22]. The aim is to provide a framework to understand nuances that shape an individual’s personal construct of dyspnoea and offer practical suggestions to challenge unhelpful beliefs and facilitate cognitive restructuring as a pathway to reduce distress and optimise health behaviours and outcomes. This is important as the skills needed to facilitate these interactions may be lacking as healthcare professionals often report low levels of confidence and limited formal training in the implementation of psychological interventions [23].

## 2. Using the Common-Sense Model to Understand an Individual’s Perception of Dyspnoea

A person’s expectations and beliefs are major determinants of their health behaviours [18]. According to the CSM, when a person starts to experience dyspnoea on exertion, they attempt to make sense of it using a range of information sources (known as situational stimuli), such as memories of prior illnesses and interactions with healthcare professionals, lay people and the media/internet [18,21]. Their personal construct of dyspnoea will be shaped by domains of identity, consequences, causes, timeline, control and coherence [18]. Thoughts around these domains will be informed by their direct and vicarious experiences that are filtered through unique social and cultural contexts [18,21]. Table 1 provides definitions of these domains and examples of questions the person may ask themselves to create a personal construct of their dyspnoea. Whilst acknowledging that people will have varied situational stimuli, Table 1 also provides examples of beliefs and expectations around dyspnoea that have been previously reported [19,20] or have been commonly encountered as part of our own clinical and research experience working with people with COPD. In contrast to other common unpleasant symptoms, such as pain, the development of a personal construct of dyspnoea is likely to be influenced by the societal and self-stigma associated with this sensation. Dyspnoea meets all six criteria of stigma: it is not concealable, is progressive, is disruptive to social situations, is aesthetically displeasing (seeing someone experiencing dyspnoea is confronting and distressing), may be perceived as harmful to others (coughing in public is a social taboo in the COVID-19 era), and is supposedly self-inflicted [24]. The stigma around dyspnoea devalues the worth of the person experiencing it, and the personal construct may therefore be influenced by emotions of shame and guilt. The CSM emphasises that the personal construct around a sensation can change based on the presentation of new information as well as experiences in response to interventions [18]. Therefore, Table 1 also presents ways to thoughtfully challenge the unhelpful beliefs that underpin the individual’s personal construct of dyspnoea. Further insight into addressing unhelpful beliefs has been addressed in the Breathing, Thinking, Functioning model developed by the Cambridge Breathlessness Intervention Service [25].

## 3. Using the Common-Sense Model to Optimise Engagement with Pulmonary Rehabilitation Programs

One way the CSM can be applied to optimise health outcomes for people with COPD is to encourage action-oriented coping strategies, such as participation in pulmonary rehabilitation programs (PRPs). It is well established that for people with COPD, PRPs, which include supervised exercise training, reduce dyspnoea and increase exercise tolerance [26]. In fact, on completion of the most recent Cochrane review, which synthesised data from 65 randomised controlled trials of PRPs, the editorial board chose to close the review, concluding “…those who apply the intervention, those who receive it, and those who fund it can act with confidence [27]”.

Despite the evidence, implementation of these programs is problematic [28]. A recent study completed by our group demonstrated that the proportion of people with COPD who were both eligible and appropriate for referral to a PRP who actually went on to be referred to this intervention was only 47% [20]. Even fewer people will complete a program [29]. Efforts to address this issue have focused on overcoming practical matters, such as difficulties with transport and parking costs, or the inflexibility of program availability, leading to difficulties fitting it in around work or career commitments. This has resulted in studies exploring new models of delivery to improve access, such as telerehabilitation and low-cost, home-based models [30]. Little attention has been given to unhelpful beliefs and the associated emotional responses to dyspnoea (e.g., distress, fear or anxiety), which may result in low uptake of referrals and high program attrition. Our earlier study identified that those who were appropriate for a referral to a PRP but did not go on to be referred were characterised by having less interest in a program [20]. When probed about their lack of interest, people expressed unhelpful beliefs that shaped their personal construct of dyspnoea and described these as barriers to accepting a referral to a PRP. Using the CSM, the interaction between these unhelpful beliefs, avoidant coping strategies and health outcomes is depicted in Figure 1.

## 4. The Importance of Helping People to “Make Sense” of Their Dyspnoea

Sensations that are perceived as unpredictable and uncontrollable are often interpreted as having harmful and even life-limiting consequences [21]. Perceiving dyspnoea as unpredictable [31] exacerbates emotional distress and panic [32] and is likely to result in avoidant coping strategies, such as denial and emotional venting [18]. Poor nighttime sleep quality and daytime sleepiness is likely to exacerbate avoidant coping strategies and distress [33,34]. The idea that dyspnoea evokes a strong emotional reaction and induces changes in behaviour has been coined the “dyspnoea–inactivity vicious cycle” [35], with dyspnoea-related kinesiophobia being a pervasive feature of COPD [36]. Indeed, if people with COPD believe that their dyspnoea on exertion is dangerous and something to be feared, then avoidance of activity is a common-sense problem-solving approach to minimise harm to their body [18,21].

Challenging unhelpful beliefs (e.g., dyspnoea is dangerous and should be avoided) is likely to be difficult as people with COPD have often lived with their dyspnoea for several decades, so their personal construct of this sensation is likely to be strongly held. It is also likely to be reinforced through confirmatory bias as people who are socially connected to the person with dyspnoea (e.g., partners, adult children, and neighbours) may hold similar unhelpful beliefs [37]. Simply observing dyspnoea in others induces mild-to-moderate dyspnoea in the observer and evoke negative affective responses [38]. In this way, those socially connected with the person with COPD often reinforce avoidant coping strategies to minimise their own distress [37,38]. To be successful in challenging these unhelpful beliefs, a strong therapeutic alliance is required. Communication should be open, empathetic and reflective, and people who are socially connected to the person with COPD, especially those who hold similar unhelpful beliefs, should be included in these therapeutic conversations. With increased trust, people can be prompted to reflect on experiences and influences that led to these misconceptions and the impact these beliefs have on their behaviour [22].

In addition to addressing unhelpful beliefs that dyspnoea is dangerous, fear is also likely to stem from the incongruence that they are experiencing worsening dyspnoea despite the reduction in physical activity (i.e., “it is getting worse even though I am taking it easy”) [21]. That is, if people hold the belief that “the only way to reduce my dyspnoea is to rest” but continue to experience worsening dyspnoea despite withdrawing from most physical activity, the discrepancy between the expected and actual outcomes of the behaviour is confusing and frightening [22]. This can be allayed by explaining to the person that whilst reducing physical activity during daily life minimises the experience of dyspnoea-related fear in the short term, it results in unwanted sequelae, such as cardiovascular and skeletal muscle deconditioning, which, paradoxically, contributes to experience of dyspnoea over and above any impairment in lung function [39]. Encouraging people to reflect on the incongruence between the protective behaviour (rest) and desired outcome (less dyspnoea) will challenge this misconception. The key concepts to challenge and unpack in order to create a new nonthreatening personal construct of dyspnoea are summarised in Table 2. Using the CSM, Figure 2 depicts the way in which challenging unhelpful thoughts and beliefs can help the person make sense of their sensation to facilitate action-orientated coping strategies, such as participation in PRPs [18].

Although important, therapeutic conversation to address unhelpful beliefs alone is unlikely to be sufficient to produce a sustainable change in people’s cognitive representation of dyspnoea. In the pain literature, in addition to therapeutic conversations, therapeutic interventions, such as behavioural experiments, designed to violate expectations are needed to facilitate cognitive restructuring [40]. In COPD, this might look like asking the person to engage in small amounts of physical activity and adopt strategies to relieve dyspnoea on exertion (e.g., pursed lip breathing, forward leaning positioning, distraction, reassurance and fans) and monitor them during recovery following exertion. In our experience, it is helpful to ask someone with COPD how far they can walk before their breathlessness stops them and also to predict what would happen if they continued to try and walk once their dyspnoea had stopped them. After this, we ask them to perform a six-minute walk test (6MWT) and compare the actual results with the predicted results. For example, a person might tell the healthcare professional that they can walk for one or two minutes and cover less than 100 m before they need to stop and if they recommenced walking they fear a catastrophic outcome. When they complete a 6MWT in the presence of a calm, reassuring healthcare professional, they often do much better than they predicted. They may choose to stop and rest during the test but can often recommence walking and cover more ground than expected. It is surprising to many that the recovery time on test completion is often less than 2 min and that there are no harmful effects of feeling this intensity of dyspnoea on exertion. Although being empathetic to any distress reported by the person with COPD is important, it is essential that healthcare professionals themselves do not show any distress. These mastery experiences violate their beliefs that dyspnoea is dangerous and unpredictable.

## 5. Evidence That This Approach Reduces the Activity Restriction Associated with Dyspnoea

Although this approach of cognitive restructuring may appear to be unfamiliar to healthcare professionals who work with people who are distressed by dyspnoea, we contend this process underpins many of the gains seen following a PRP. We question the likelihood that, in routine clinical practice, the reduction in dyspnoea and improvement in exercise tolerance seen on completion of exercise training is due solely to the commonly ascribed mechanism of improved oxidative capacity of the quadricep muscles [26]. This is because participants in the seminal studies who demonstrated improved oxidative capacity of the quadriceps with exercise training completed a program that was far more intense than is common in usual clinical practice [41,42,43]. This was possible, at least in part, because participants in these earlier studies had minimal, if any, comorbidity. In contrast, people referred to clinical PRPs have complex health needs, with nearly 30% having five or more comorbid conditions [44]. Also, large reductions in dyspnoea have been shown on completion of exercise training programs performed at intensities that are unlikely to be sufficient to produce important changes in the oxidative capacity of the skeletal muscle [45]. For these reasons, we believe that the reduction in dyspnoea and improved exercise tolerance seen on completion of a clinical PRP is largely because the exercise training component has served as a behavioural experiment (akin to exposure therapy) [21]. During a PRP, people with dyspnoea on exertion are supported to progressively increase their participation and tolerance of their perceived threatening activity (exercise). Over time, regular participation in exercise disconfirms their unhelpful beliefs that their dyspnoea is dangerous. Reductions in fear associated with dyspnoea will increase self-efficacy for exercise, which is a primary determinant of change in exercise capacity in response to a PRP [46]. These graded behaviour experiments when accompanied by explicit therapeutic conversations are likely to be a powerful way to help the person develop a new nonthreatening cognitive representation of their dyspnoea while adopting positive health behaviours. Indeed, in people with chronic low back pain, where unhelpful beliefs, emotions and behaviour are drivers of pain and disability, this approach, known as cognitive functional therapy, has been shown to produce clinically important reductions in pain and activity limitation that were sustained for at least 12 months [40,47].

## 6. Conclusions

The CSM can be used to shape an individual’s personal construct of dyspnoea and provide suggestions to challenge unhelpful beliefs and facilitate cognitive restructuring through sense-making processes and behavioural experimentation. Discussions that seek to understand and challenge unhelpful beliefs should be embedded in PRPs, especially those characterised by high levels of fear related to the person’s experience of dyspnoea. Of particular importance, given the poor implementation and uptake of PRPs, studies are needed to explore the impact that addressing unhelpful beliefs has on accepting a referral to these services. It is possible that helping the person make sense of their sensation and creating a new nonthreatening personal construct of dyspnoea may optimise the number of people who feel safe to participate in, and benefit from, PRPs.

## Figures and Tables

**Figure 1 jcm-13-00200-f001:**
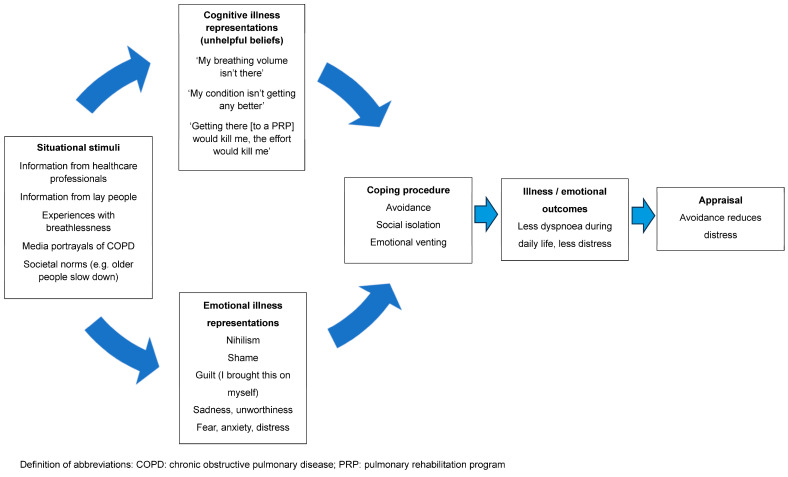
A representation of the interaction between unhelpful beliefs, avoidant coping strategies and health outcomes.

**Figure 2 jcm-13-00200-f002:**
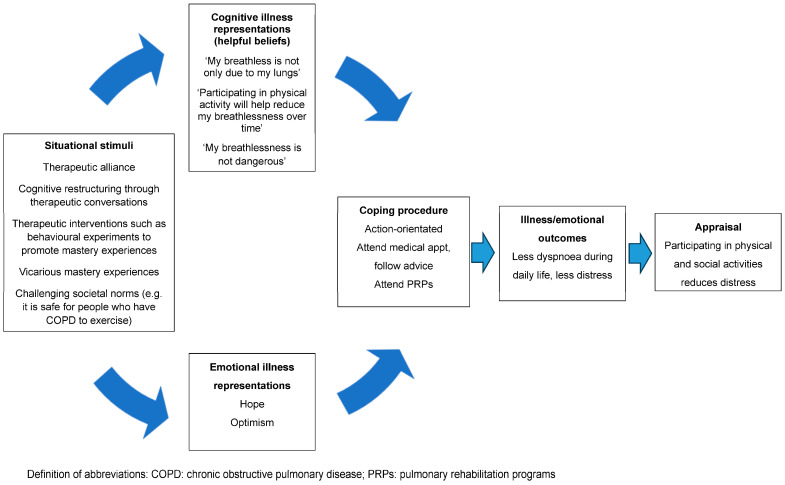
A representation of the way challenging unhelpful thoughts and beliefs may help the person make sense of their dyspnoea to facilitate action-orientated coping strategies.

**Table 1 jcm-13-00200-t001:** Examples of the ways people with COPD develop a personal construct of dyspnoea.

Domains	Probing Questions the Person Will Ask to Develop Their Construct	Common Thoughts and Beliefs for People with COPD about Their Dyspnoea	Media and Social Influences Related to Dyspnoea in COPD	Ways to Challenge Unhelpful Beliefs
***Identity:*** Disease label and somatic representation of that disease	What is this sensation?	*The sensation is highly variable*. * It was initially attributed to ageing and weight gain. *	Lung disease is invisible, and the symptoms are isolating/stigmatising. *	The limitation caused by breathlessness is not strongly related to lung function—there are many other factors that contribute to this sensation.
***Cause:*** Antecedents	What caused the sensation?	*Breathlessness is caused by my lung disease, which is because I smoked for several years.* *I get breathless whenever I exert myself.* *My oxygen levels must be low*.	There is very little societal empathy for smoking-related health conditions (they could be avoided if the person did not smoke). In movies, putting supplemental oxygen on people who are breathless quickly alleviates the symptom.	Most people who smoke do not develop COPD. In most people with COPD, breathlessness is not caused by low oxygen levels.
***Consequences:*** Anticipated repercussions	What will be the consequences of this sensation?	*I can no longer participate in valued life activities*. * *I feel like a burden on others.* * *My breathlessness is harmful and damages my body*. *If I get very breathless, I might have a heart attack and die.* *I need to call an ambulance*. *I may lose control of my bladder and/or bowel when I am breathless and worry I will not make it to the bathroom on time*.	Family members find witnessing breathlessness in their loved one as frightening, so they discourage them from undertaking certain activities that bring on their breathlessness.	Breathlessness on exertion in people with COPD is not harmful and is very rarely life threatening.
***Control:*** Responsiveness to intervention	How can I control this sensation?	*If I am outside and cannot find somewhere to sit down, I will panic.* Medical treatment is going to reduce my breathlessness so that I can get back to feeling like I did 10 years ago. Factors used to control dyspnoea include rest, stopping, pacing distraction, exercise, avoiding triggers, self-talk and relaxation. *	Healthcare providers rarely ask people about their breathlessness, so maybe nothing can be done about it.	Exercise, when prescribed and monitored by a qualified healthcare professional, is a safe and an effective way to learn to cope with your breathlessness.
***Timeline:*** Acute, chronic or cyclic	How long will this sensation last?	*The future is uncertain and unpredictable*. * *There is no cure for emphysema*. *The sensation will be with me for the rest of my life*. *Once diagnosed, I will only get worse (especially if they have experienced someone close to them (e.g., their father) succumb to their COPD)*.	It is a progressive disease.	Once you quit smoking, the rate of decline in lung function returns to that expected due to the ageing process. Dyspnoea varies both within and across days.
***Coherence:*** Making sense	What do I understand about this sensation?	*Measurements made by healthcare professionals (e.g., oximetry and spirometry) do not convey the distress my breathlessness causes me. **	People who are breathless are often told to “not overdo it” or “take it easy”.	You have avoided activity for a long time and yet your breathlessness continues to get worse. Why is that?

* Insights previously reported [19]. Comments in italics are likely to be associated with an unpleasant emotional response, such as distress, fear and anxiety, and lead to avoidant coping strategies.

**Table 2 jcm-13-00200-t002:** Examples of therapeutic conversations to challenge unhelpful beliefs.

Threshold Concept	Explanation	New Beliefs
Dyspnoea will be frightening.	The brain perceives dyspnoea as a risk to your survival. It activates structures in your brain that give rise to fear.	The perception of fear and the true threat to your survival are not closely linked (for example, nightmares, social anxiety, most phobias, etc).
Chronic dyspnoea is rarely dangerous.	The brain does not separate acute (dangerous) dyspnoea from chronic (unpleasant but rarely dangerous) dyspnoea.	Explain the difference between acute dyspnoea, which can be dangerous, and chronic dyspnoea, which is unlikely to be dangerous.
Avoiding activity is a common-sense approach to reduce chronic dyspnoea.	Because you perceive it is dangerous, this will change your behaviour and you will find ways to avoid it (self-preservation).	You are not avoiding activity because you are “lazy”. Feeling breathless is frightening and often perceived as causing harm to our body, so it makes sense to avoid it.
Avoiding activity is reinforced in the short term.	This strategy is reinforced because if you avoid physical activity, you feel less dyspnoea during everyday life.	Short-term strategies may have short-term benefits but will result in long-term decline.
Avoiding activity does not work in the long term and your dyspnoea will gradually worsen.	Avoiding physical activity over several months or years weakens your heart and leg muscles, and these processes contribute to your sense of dyspnoea (independent of lung function).	How has your dyspnoea changed over the last few years? Is avoiding activity making it better in the long term?
Improvements will not be linear.	Returning to activity is challenging. It is not like taking an antibiotic, where each day you will get a little bit better. You will continue to have good days and bad days.	It is a chronic condition, and exacerbations are inevitable.
There are strategies that can help.	Returning to physical activity will not change your lung function, but it will reduce your breathlessness.	Address nihilism, hopelessness and helplessness (with mastery experiences).

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
