# Peer review of "The Role of Illness Perceptions in Dyspnoea-Related Fear in Chronic Obstructive Pulmonary Disease"

_jcm, 2023, doi:10.3390/jcm13010200_

Round 1

Reviewer 1 Report

Comments and Suggestions for Authors

The authors should be congratulated for the topic. Recently, the interest in mental health burden and respiratory disease such as OSAS or COPD are emerging and represents a hot topic to discuss. The manuscript added important evidence to the previous literature such as practical suggestions to facilitate cognitive structuring, and optimizing health behaviors. I suggest discussing also this novel and recent paper (PMID: https://doi.org/10.3390/jcm12226965). The authors measured high levels of psychological distress in people who experienced both lower urinary tract symptoms and excessive daytime sleepiness (related to OSAS conditions). Specifically, it must be acknowledged that a condition of respiratory impairment is severely related to the mental health burden (PMID= 37647715). Moreover, I think that the captions of Figures 1 and 2 are lost. They should be added to better understand the figure diagrams. 

Author Response

Reviewer 1

Comment 1: Mental health burden and respiratory disease such as OSAS or COPD are emerging and represents a hot topic to discuss. The manuscript added important evidence to the previous literature such as practical suggestions to facilitate cognitive structuring, and optimizing health behaviors. I suggest discussing also this novel and recent paper (PMID: https://doi.org/10.3390/jcm12226965). The authors measured high levels of psychological distress in people who experienced both lower urinary tract symptoms and excessive daytime sleepiness (related to OSAS conditions). Specifically, it must be acknowledged that a condition of respiratory impairment is severely related to the mental health burden (PMID= 37647715).

Response 1: We now comment on the relationship between poor nighttime sleep quality and/or daytime sleepiness and avoidant coping strategies. We have added the references as suggested.

Comment 2: Moreover, I think that the captions of Figures 1 and 2 are lost. They should be added to better understand the figure diagrams. 

Response 2: Thank-you. These have now been added.

Reviewer 2 Report

Comments and Suggestions for Authors

Manuscript ID: JCM-2716994

Reviewer Report

Thank you for giving me a chance to review this study. It is a good review of

The role of illness perceptions in dyspnoea-related fear in chronic obstructive pulmonary disease

Dear author,

Comments

1.     Abstract: Should be structured. It should have a background, Method, Results, Conclusion.

2.     Introduction:   Line no 69 This model has been recently revised (15). This sentence is very short and no connection. Please explain in detail

3.     Few more references could be added to support since it is a review.

4.     Include more details about the pathology of COPD and its association with dyspnea. Separate section is needed.

5.     Method section?

6.     Authors did not mention about the type of review. Narrative review?

If so. Under methods: need to mention the data based used and type of articles selected for this study and year and language, filters used. I suggest author to read narrative review article and add the details accordingly.

7.     Overall I feel very different review and authors used Common Sense Model to understand an individual’s perception of dyspnoea and it is highly appreciated.

Best wishes

Best Wishes

Author Response

Reviewer 2

Comment 1: Abstract: Should be structured. It should have a background, Method, Results, Conclusion

Response 1: We followed the instructions for authors for the JCM that state:

The abstract should be a total of about 200 words maximum. The abstract should be a single paragraph and should follow the style of structured abstracts, but without headings: 1) Background: Place the question addressed in a broad context and highlight the purpose of the study; 2) Methods: Describe briefly the main methods or treatments applied. Include any relevant preregistration numbers, and species and strains of any animals used; 3) Results: Summarize the article's main findings; and 4) Conclusion: Indicate the main conclusions or interpretations. The abstract should be an objective representation of the article: it must not contain results which are not presented and substantiated in the main text and should not exaggerate the main conclusions.

According to these instructions, it appears that no headings within the abstract are permitted.

Comment 2: Introduction: Line no 69 This model has been recently revised (15). This sentence is very short and no connection. Please explain in detail

Response 2: We have now added some detail to clarifying the nature of this revision.

Comment 3: Few more references could be added to support since it is a review.

Response 3: We added 6 more references (all published between 2019 and 2023).

Comment 4: Include more details about the pathology of COPD and its association with dyspnea. Separate section is needed.

Response 4: Thank-you for this suggestion. We do not want to emphasise the pathophysiology of COPD and dyspnoea which serves to strengthen the biomedical model. Therefore, in response to this comment we have chosen to add some additional information, but not an entire separate section.  

Comment 5: Method section?

Response 5: This paper was submitted as a ‘Review’ (not a systematic review) and we followed the instructions for authors for this submission type – which are as follows:

Review

Reviews offer a comprehensive analysis of the existing literature within a field of study, identifying current gaps or problems. They should be critical and constructive and provide recommendations for future research. No new, unpublished data should be presented. The structure can include an Abstract, Keywords, Introduction, Relevant Sections, Discussion, Conclusions, and Future Directions, with a suggested minimum word count of 4000 words.

According to these instructions, it appears that no methods section was required.

Comment 6: Authors did not mention about the type of review. Narrative review? If so. Under methods: need to mention the data based used and type of articles selected for this study and year and language, filters used. I suggest author to read narrative review article and add the details accordingly.

Response 6: We have now made it clear this is a narrative review (that draws on earlier work that has applied the CSM to understand beliefs and expectations around dyspnoea, as well as our own clinical and research experience working with people with COPD). Also, please see response to comment 5.

Comment 7: Overall I feel very different review and authors used Common Sense Model to understand an individual’s perception of dyspnoea and it is highly appreciated.

 Response 7: Many thanks for this kind comment.

Reviewer 3 Report

Comments and Suggestions for Authors

Indeed, dyspnea is a very limiting and distressing symptom for patients with chronic respiratory diseases. Likewise, perceived dyspnea may be conditioned by other factors as indicated in the submitted manuscript, congratulations! Although it is very interesting for potential readers of the Journal, some comments are made in order to improve the current version of the manuscript: .- Títle. Questions arise regarding the use of the word “fear”. The manuscript also talks about anguish and panic. Maybe it would be better to include the term “emotions” in the title? The basic emotions include sadness. Why was it not included as a review topic?  .- Abstract, ok. .- Keywords. Maybe “emotions” instead of “emotion”. Consider adding “perceptions”, “health behaviors” and “coping strategies” .- Introduction, ok. .- Subsections 2-5, ok. .- Conclusions. Maybe “In this paper, we introduce a way in which” could be removed.  The acronym “PRP” is repeated up to four times. Review. .- Tables and figures. Two tables and two figures. Figure 2. Blurred text. Review. Table 2. Maybe “Examples” is better. .- References. 41 citations are provided, of which twelve (29.2%) are recent (that is, five years or less since their publication). It would be appreciated by the reader given the importance, relevance and breadth of the topic (perceived dyspnea, emotional response, PRP, etc.) to include some additional citation, if possible.  Consider including any additional recent references.

Author Response

Reviewer 3

Comment 1: Indeed, dyspnea is a very limiting and distressing symptom for patients with chronic respiratory diseases. Likewise, perceived dyspnea may be conditioned by other factors as indicated in the submitted manuscript, congratulations! Although it is very interesting for potential readers of the Journal, some comments are made in order to improve the current version of the manuscript.

Response 1: Thank you for this kind comment. We are excited to have this paper published.

Comment 2: Títle. Questions arise regarding the use of the word “fear”. The manuscript also talks about anguish and panic. Maybe it would be better to include the term “emotions” in the title? The basic emotions include sadness. Why was it not included as a review topic?  

Response 2: We understand and have carefully considered the reviewer’s comment. However, we feel that fear is the most pervasive symptom that appears to underpin the adoption of avoidant coping strategies, and we have added references to support this view. Because of this, we would prefer to keep the term ‘fear’ in the title. Nevertheless, we have now added ‘sadness’ as an important consideration within Figure 1. We also mention hopelessness and helplessness in Table 2, which are hallmarks of depression.

Comment 3: Abstract, ok. 

Response 3: Thank-you.

Comment 4: Keywords. Maybe “emotions” instead of “emotion”. Consider adding “perceptions”, “health behaviors” and “coping strategies” 

Response 4: We have changed ‘emotion’ to ‘emotions’ and added ‘perceptions, health behaviours and coping strategies’ as suggested.

Comment 5: Introduction, ok. .- Subsections 2-5, ok. .- Conclusions. Maybe “In this paper, we introduce a way in which” could be removed.  The acronym “PRP” is repeated up to four times. 

Response 5: We have modified the conclusion as suggested and also reduced the use of the abbreviation (PRP).

Comment 6: Review. .- Tables and figures. Two tables and two figures. Figure 2. Blurred text. Review. Table 2. Maybe “Examples” is better. .- 

Response 6: Changes made as suggested (also figures have been re-formatted so that they are no longer blurred).

Comment 7: References. 41 citations are provided, of which twelve (29.2%) are recent (that is, five years or less since their publication). It would be appreciated by the reader given the importance, relevance and breadth of the topic (perceived dyspnea, emotional response, PRP, etc.) to include some additional citation, if possible.  Consider including any additional recent references.

Response 7: We have added 6 new references, all of which have been published between 2019 and 2023.

Reviewer 4 Report

Comments and Suggestions for Authors

The role of illness perceptions in dyspnoea-related fear in chronic obstructive pulmonary disease

The current article describes a review pertaining to dyspnoea related construct in patients with chronic obstructive pulmonary disease. Dyspnoea is one of the most important markers utilized in clinical settings when trying to determine severity of illness and functional compromise among patients. Like authors mentioned, it is associated with strong emotional responses both from patients and family. Most of the self-efficacy surrounding management of dyspnoea in this group of patients depends on various clinical and non-clinical factors.

Traditional model describes dyspnoea as a neuromechanical uncoupling seen in setting of dynamic hyperinflation frequently noted among these patients. However, authors debate that there are various other factors which discriminate a given patient’s response to the occurrence and progression of this symptom. Here is where they introduce the Commonsense Model to identify individual’s perception of this symptom. Table 1 describes various experiences and their influence on a person’s personal construct towards dyspnoea. Changing patient’s beliefs by challenging with a positive affirmation is demonstrated. Figure 1 also illustrates howe we can provide the right direction during coping procedures or mechanisms for optimal response from patients.

By improving understanding of their personal symptoms, and by providing healthy coping strategies, there is an argument for improving engagement in pulmonary rehabilitation centers. This also pays forward in management of this chronic disease by making long-term strides in their health. Finally, the article provides evidence supporting the statement that improvement in exercise tolerance and dyspnoea seen after rehabilitation cannot be attributed solely to oxidative capacity improvement of quadriceps muscle only. It's mostly due to the change in inner construct and beliefs of patients and this change in understanding of their symptoms that lead to positive outcomes.

Author Response

Reviewer 4

Comment 1: The current article describes a review pertaining to dyspnoea related construct in patients with chronic obstructive pulmonary disease. Dyspnoea is one of the most important markers utilized in clinical settings when trying to determine severity of illness and functional compromise among patients. Like authors mentioned, it is associated with strong emotional responses both from patients and family. Most of the self-efficacy surrounding management of dyspnoea in this group of patients depends on various clinical and non-clinical factors. Traditional model describes dyspnoea as a neuromechanical uncoupling seen in setting of dynamic hyperinflation frequently noted among these patients. However, authors debate that there are various other factors which discriminate a given patient’s response to the occurrence and progression of this symptom. Here is where they introduce the Commonsense Model to identify individual’s perception of this symptom. Table 1 describes various experiences and their influence on a person’s personal construct towards dyspnoea. Changing patient’s beliefs by challenging with a positive affirmation is demonstrated. Figure 1 also illustrates howe we can provide the right direction during coping procedures or mechanisms for optimal response from patients.

By improving understanding of their personal symptoms, and by providing healthy coping strategies, there is an argument for improving engagement in pulmonary rehabilitation centers. This also pays forward in management of this chronic disease by making long-term strides in their health. Finally, the article provides evidence supporting the statement that improvement in exercise tolerance and dyspnoea seen after rehabilitation cannot be attributed solely to oxidative capacity improvement of quadriceps muscle only. It's mostly due to the change in inner construct and beliefs of patients and this change in understanding of their symptoms that lead to positive outcomes.

Response 1: Many thanks for this thoughtful synopsis of our work.

Round 2

Reviewer 2 Report

Comments and Suggestions for Authors

Comment 6: Authors did not mention about the type of review. Narrative review? If so. Under methods: need to mention the data based used and type of articles selected for this study and year and language, filters used. I suggest author to read narrative review article and add the details accordingly.

Not addressed well. As this review is narrative  review defnitely authors should address the above comment. 

Author Response

We apologise for the confusion. On page 4 of the revision (word document), we had added the word 'narrative' - i.e.

'In this narrative review, we draw on earlier work.....'   

When preparing this manuscript, we carefully read the Instructions for Authors for JCM - and for a review, these are as follows:

'Review: Reviews offer a comprehensive analysis of the existing literature within a field of study, identifying current gaps or problems. They should be critical and constructive and provide recommendations for future research. No new, unpublished data should be presented. The structure can include an Abstract, Keywords, Introduction, Relevant Sections, Discussion, Conclusions, and Future Directions,'

According to these instructions, there is no need for a methods section.

This is in contrast with a systematic review - for which the 'Instructions for Authors' state:

'Systematic Review: Systematic review articles present a detailed investigation of previous research on a given topic that use clearly defined search parameters and methods to identify, categorize, analyze, and report aggregated evidence on a specific topic. The structure is similar to a review, with a suggested minimum word count of 4000 words; however, they should include a Methods section.'

So to clarify, the paper we have prepared is a review, not a systematic review. As such, in accordance with the Instructions for Authors on the JCM website, there is no methods section (as we did not undertake a systematic review of the literature). Instead, as stated in the manuscript, 'we draw on earlier work that has applied the CSM to understand beliefs and expectations around dyspnoea [19], as well as our own clinical and research experience working with people with COPD [20].' 

We this addresses the Reviewer's concerns.